# Parkinson’s Disease Gene Biomarkers Screened by the LASSO and SVM Algorithms

**DOI:** 10.3390/brainsci13020175

**Published:** 2023-01-20

**Authors:** Yiwen Bao, Lufeng Wang, Fei Yu, Jie Yang, Dongya Huang

**Affiliations:** Tongji University School of Medicine, East Hospital, Department of Neurology, Tongji University, Shanghai 200070, China

**Keywords:** Parkinson’s disease, immune infiltrates, least absolute shrinkage and selection operator, support vector machine

## Abstract

Parkinson’s disease (PD) is a common progressive neurodegenerative disorder. Various evidence has revealed the possible penetration of peripheral immune cells in the substantia nigra, which may be essential for PD. Our study uses machine learning (ML) to screen for potential PD genetic biomarkers. Gene expression profiles were screened from the Gene Expression Omnibus (GEO). Differential expression genes (DEGs) were selected for the enrichment analysis. A protein–protein interaction (PPI) network was built with the STRING database (Search Tool for the Retrieval of Interacting Genes), and two ML approaches, namely least absolute shrinkage and selection operator (LASSO) and support vector machine recursive feature elimination (SVM-RFE), were employed to identify candidate genes. The external validation dataset further tested the expression degree and diagnostic value of candidate biomarkers. To assess the validity of the diagnosis, we determined the receiver operating characteristic (ROC) curve. A convolution tool was employed to evaluate the composition of immune cells by CIBERSORT, and we performed correlation analyses on the basis of the training dataset. Twenty-seven DEGs were screened in the PD and control samples. Our results from the enrichment analysis showed a close association with inflammatory and immune-associated diseases. Both the LASSO and SVM algorithms screened eight and six characteristic genes. AGTR1, GBE1, TPBG, and HSPA6 are overlapping hub genes strongly related to PD. Our results of the area under the ROC (AUC), including AGTR1 (AUC = 0.933), GBE1 (AUC = 0.967), TPBG (AUC = 0.767), and HSPA6 (AUC = 0.633), suggested that these genes have good diagnostic value, and these genes were significantly associated with the degree of immune cell infiltration. AGTR1, GBE1, TPBG, and HSPA6 were identified as potential biomarkers in the diagnosis of PD and provide a novel viewpoint for further study on PD immune mechanism and therapy.

## 1. Introduction

Parkinson’s disease (PD) is a common progressive neurodegenerative disorder. Its main pathological features are the loss of dopaminergic neurons in the substantia nigra (SN) and α-synuclein abnormal aggregation [1]. PD is usually diagnosed by the physical examination and assessment of motor symptoms, such as resting tremor, muscle rigidity, and bradykinesia [2].

Currently, the treatment for PD is mainly to relieve symptoms by adding levodopa, dopamine receptor agonists, etc. These treatments can control motor symptoms only in the early stages of PD and do not prevent dopaminergic neuronal damage. As the disease progresses and the response to the drugs decreases, long-term use of these drugs is accompanied by adverse effects, such as dyskinesia and symptom fluctuations [3], which seriously affect the quality of life of patients [4]. Therefore, better and more effective therapeutic strategies are needed, and the fundamental step toward this goal is to search for the underlying genetic and molecular mechanisms behind the pathogenesis of PD.

Increasing research indicates that both innate and adaptive immune responses play a pivotal role in the pathogenesis of PD [5]. The protein α-synuclein, considered the central component to the pathogenesis of PD, was associated with the immune responses triggered by the immune cells [6]. Thus, the immunosuppressants which immunologically restore the brain’s homeostatic environment have been proven to affect the progress of PD [7]. Racette et al. performed a population-based case–control study that included 10,619 participants and found that using immunosuppressants, such as corticosteroids and inosine monophosphate dehydrogenase inhibitors, might decrease the risk of PD [8]. Peter et al. also found that early exposure to anti-tumor necrosis factor could significantly reduce the incidence of PD in patients with inflammatory bowel disease [9]. Similarly, a recent national case–control study from Finland showed that using immunosuppressants helped reduce the risk of Parkinson’s disease in rheumatoid arthritis [10]. These studies suggest the essential role of immunosuppressants in PD, and, on the other hand, they indicate that neuroinflammation is an important pathological feature in the pathogenesis of PD. It was shown that neuroinflammation is regulated by immune cells, such as microglia (macrophages), astrocytes, and peripheral immune cells, as well as cytokines [11], of which microglia play a significant mediating role [12], and the accumulation of α-synuclein is also associated with microglia activation. A considerable infiltration of CD4 and CD8 cells was observed in the postmortem NS cells of patients, and it was shown that in the absence of T cells, α-synuclein could not upregulate the microglial proinflammatory response, and there is no loss of neurons; thus, T-cell infiltration is necessary for neuronal degeneration [13]. The activation of neuroinflammatory responses by these immune cells promotes the onset of neurotoxicity, which, in turn, leads to neuronal death. To sum up, these potential immune cell infiltrations can influence the pathogenesis of PD and are also potential targets for developing PD-modifying therapies [14].

With the rapid development of bioinformatics, compared with time-consuming and expensive traditional experimental research, the bioinformatics analysis can screen a larger number of potentially worthwhile genes more quickly and accurately and provide exploratory predictions at a lower cost to inform subsequent biological experiments and clinical applications [15]. Hub gene, as a gene with a high degree of connectivity in the gene expression network, is considered to play a pivotal role in the progression of the disease [16]. In previous studies, cytoHubba or STRING (Search Tool for the Retrieval of Interacting Genes) software was often used to screen for hub genes [17,18]. However, in this type of selection, whether to select the top 5 or 10 of total differential expression genes (DEGs) as hub genes depends on the researchers’ preferences, which inevitably decreases the accuracy of screening process and reduces the repeatability of the experiment [19,20]. In order to diminish this type of inaccuracy, various machine learning (ML) techniques have been recently added to bioinformatics analysis, which has been proven to give the screening method better accuracy and stability [21,22]. The least absolute shrinkage and selection operator (LASSO) regression, as a normalized linear regression method, can ignore unimportant features and build a sparse and easy-to-interpret model to prevent overfitting. The support vector machine recursive feature elimination (SVM-RFE) technique integrates the support vector machine into the recursive feature elimination strategy and uses the inherent feature selection function of the support vector machine to screen key features in continuous iteration. The combination of LASSO and SVM-RFE techniques has shown satisfactory accuracy and sensitivity in some fields, such as lung and pituitary tumors [23,24]. However, few studies have screened PD-related bioinformation by the combination of LASSO and SVM-RFE ML techniques.

In this study, we creatively took advantage of the combination of these two ML techniques to identify the hub genes for PD and further analyzed these gene-related infiltration patterns of PD immune cells. This combination of these two machine learning techniques could not only screen the genes with significant features but also delete the gene that has the least influence on the pathogenesis of PD. We hope our study can reveal the information regarding neuroimmune-related pathogenesis of PD more accurately and provide some insights into searching for the potential targets of immunotherapy for PD.

In the present work, we first combined PD microarray collections from the Gene Expression Omnibus (GEO) database to identify DEGs and perform enrichment analysis. Then, we combined two ML algorithms, LASSO regression, and SVM-RFE analysis to identify the PD-related hub genes. Next, the convolution tool cell-type identification by estimating relative subsets of RNA transcripts (CIBERSORT) was used to investigate the discrepancies between immune cells in PD pathogenesis and explore the correlation between hub genes and immune cell infiltration. Finally, another PD microarray dataset that met the inclusion criteria was used for external validation. The flow chart of the present study design is shown in Figure 1.

## 2. Materials and Methods

### 2.1. Data Processing and Differential Gene Screening

The GEO database (https://www.ncbi.nlm.nih.gov/geo/, accessed on 1 November 2022) is an international public repository of high-throughput microarray and next-generation sequence functional genome datasets created and maintained by The National Center for Biotechnology Information [25]. Almost all research-relevant gene expression assay data can be found in this database. Here, we extracted Parkinson’s-related microarray data from the GEO database with the selection criteria as per below: (1) organism is a Homo sapiens with a gene expression profile type of array expression profile; (2) the samples come from the substantia nigra; and (3) the raw data can be employed for further analysis. We selected four independent datasets: GSE7621, GSE20141, GSE20333, and GSE49036, including 31 normal controls and 47 PD samples from the GPL570 ((HG-U133_Plus_2) Affymetrix Human Genome U133 Plus 2.0 Array) and GPL201 ((HG-Focus) Affymetrix Human HG-Focus Target Array) platforms as test sets, and the GSE20164 microarray was selected to validate the results. The details of the 5 datasets are shown in Table 1.

We converted the gene probes into gene symbols using annotation files, where the average value of several probes corresponding to the same gene was measured. Log2 was used for the normalization. We integrated these microarray data as the training datasets after leveling out the discrepancies between the batches via a surrogate variable analysis (SVA) package [26]. The 2D principal component analysis (PCA) showed the inter-batch differences before and after treatment. We applied the “limma” package [27] in screening the DEGs by the criteria of the adj. *p*  <  0.05 and |log2 fold change (FC) |> 1, whereas we employed the “pheatmap” and “ggplot2” packages [28] in creating the heatmaps and volcano maps to visualize these DEGs.

### 2.2. Enrichment Analysis Method

To better understand the biological functions of DEGs, we used the “limma”, “clusterProfiler”, “org. Hs. eg. db”, and “DOSE” packages to perform the enrichment analysis, including Gene Ontology (GO), Kyoto Encyclopedia of Genes and Genomes (KEGG), and Disease Ontology (DO). The GO database consists of a set of terms that annotate the properties of genes and gene products. It can be analyzed at three levels: biological processes (BP), cellular components (CC), and molecular functions (MF) [29]. A KEGG analysis to assign DEGs to specific pathways is used as a database resource for understanding the network of advanced functional and interacting relationships in biological systems [30]. A DO analysis can identify multiple diseases associated with these DEGs [31]. We used the “ggplot2” package to visualize these enrichment analyses using adjusted *p*-values of <0.05 and q-values of <0.05 (Benjamini–Hochberg method) as default cut-off thresholds.

### 2.3. PPI Network Construction

The DEGs were imported into the online database STRING [32] (https://cn.string-db.org/, accessed on 6 November 2022), the Homo sapiens race was selected, and the interaction score was set to >0.15 to construct the protein–protein interaction (PPI) network. In this network, each node represents a target gene and the lines between the nodes represent related interactions. The main modules in the PPI network were analyzed in Cytoscape3.9.1 software [33] using the plug-in Minimal Common Oncology Data Elements (MCODE) [34], with the filtering parameters set to degree cutoff = 2, node score cutoff = 0.2, k-core = 4, and max. depth = 100.

### 2.4. Machine Learning Screening and Validation Gene Biomarkers

To diagnose and predict diseases more accurately, researchers have proposed various ML algorithms. Here, we applied two ML algorithms to screen hub genes. LASSO regression [35] uses regularization to improve the prediction accuracy, which is conducted by performing variable selection and complexity adjustment while fitting a generalized linear model. Here, selectively placing variables into the model was conducted to obtain better performance parameters, and then the complexity of the model was controlled by a series of parameters. The degree of the complexity adjustment was controlled by the parameter λ, which controls the severity of the penalty. In this study, the value of λ is determined by cross-validation using the “glmnet” package of R to fit the model, where the response type is set to “binary”, alpha = “1”, and nfold = “10”. An SVM-RFE analysis [36] was used to iteratively construct the model, and then the best features were selected with a sequential backward selection algorithm based on the maximum interval principle of SVM. In the research, the defined training model and cross-validation were used to obtain the value with the minimum error as the feature genes. The SVM classifier was performed with the “e1071”, “kernlab”, and “caret” packages. Additionally, we surveyed LASSO-SVM to screen the hub genes for PD and then used the “Venn” package to obtain two overlapping hub genes as potential biomarkers for PD. Furthermore, we confirmed the differences in the biomarkers’ expression of the candidate genes in the validated dataset GSE20164.

### 2.5. Diagnostic Value of Gene Biomarkers in PD

We applied receiver operating characteristic (ROC) curve analyses [37] to investigate the regression model, verified with an external validation dataset to determine the potential predictive value of the gene expression differences. The area under the ROC curve (AUC) was extremely close to 1, indicating good specificity and sensitivity of the screened genes, implying greater accuracy as potential biomarkers of disease. The AUC in the study >0.6 showed a relatively satisfying diagnosis efficiency.

### 2.6. Analysis of Immune Cell Components

The CIBERSORT [38] refers to a computational method quantifying the cell composition of complex tissues on the basis of the corresponding gene expression profiles, which should be able to analyze the RNA mixtures of cell biomarkers and therapeutic targets on a large scale. R’s “CIBER-SORT” package was employed for quantifying the relative ratio of 22 infiltrating immune cells. Samples with *p* < 0.05 were filtered out, with the zero-value type of immune cells excluded while obtaining the immune cell infiltration matrix. The packages “corrplot” and “vioplot” were employed to draw the bar and violin charts using the data on the immune cell infiltration matrix. This can help demonstrate the correlation and difference in immune cells in the SN of PD patients and normal controls. Finally, the correlation between the screening genes and immune cells was further validated by Pearson correlation analysis to explore how these genes regulate immune cell infiltration to influence the development of PD.

### 2.7. Statistical Analysis

We implemented all statistical analyses with R software (version 4.2.2). For the continuous variables, we used Student’s *t*-test, and for the categorical variables, we used the Mann–Whitney U test. *p* < 0.05 indicated statistical significance.

## 3. Results

### 3.1. Recognition of DEGs

The PCA cluster charts showed a random distribution of samples after the removal of the batch differences (Figure 2A,B). Compared with the control sample, we acquired 27 DEGs, including 25 upregulated and 2 downregulated genes, from the SN of PD patients. In addition, we visualized DEGs with heatmaps and volcano plots (Figure 2C,D).

### 3.2. DEGs Gene Enrichment Analysis

Through the GO enrichment analysis, we found that these genes showed their discrepancies mainly in the enrichment of BP, CC, and MF, such as neurotransmitter transport, dopamine biosynthetic process, synapse organization, presynapse, synaptic vesicle, exocytic vesicle, neuron projection terminus, and transport vesicle (Figure 3A). Furthermore, the KEGG pathways enrichment analyses showed the enrichment of DEGs under cocaine addiction (hsa05030), dopaminergic synapse (hsa04728), amphetamine addiction (hsa05031), alcoholism (hsa05034), synaptic vesicle cycle (hsa04721), serotonergic synapse (hsa04726), and tyrosine metabolism (hsa00350) (Figure 3B). The DEGs enrichment analysis for the disease group and control group indicated that PD possibly causes inflammatory and tumor problems in the nervous system. Autonomic nervous system neoplasm, neuroblastoma, peripheral nervous system neoplasm, Parkinson’s disease, and synucleinopathy were the first five differential genes enriched (Figure 3C). The above findings obtained with GO, KEGG, and DO indicate that there is an appropriate immune response mechanism in PD.

### 3.3. PPI Network Construction

A 27-DEG PPI network was built with the STRING database to investigate the interaction among robust DEGs. Confidence of >0.15 and the separated nodes were hidden, and all 25 nodes and 96 edges participated in the PPI network (Figure 4A). The PPI data were imported into the Cytoscape software, and we identified two significant modules on the basis of the filtering criteria by MCODE. Subcluster 1 had the high cluster score of 9.111, which included a total of 10 nodes and 41 edges, and Subcluster 2 had a score of 4, with 5 nodes and 8 edges. (Figure 4B,C).

### 3.4. Application of Machine Learning and Validation of Candidate Gene Biomarkers

To extract PD-related gene biomarkers from DEGs, we used the LASSO model and SVM-RFE. With the help of LASSO regression, eight genes were mined (Figure 5A). Six characteristic genes were degraded for the SVM-RFE algorithm (Figure 5B). At the same time, four overlapping genes were found, namely angiotensin II type 1 receptor (AGTR1), glycogen branching enzyme (GBE1), trophoblast glycoprotein (TPBG), and heat shock 70-kDa protein 6 (HSPA6) (Figure 5C).

Using the validation dataset, we further clarified its validity by verifying the expression level of the characteristic genes. As seen from the boxplots results, in GSE20164, the expression degrees of AGTR1 and GBE1 in the PD group were significantly lower than those in the control group (*p* < 0.05), while the TPBG expression was lower and the HSPA6 expression was higher without statistical significance (Figure 6).

### 3.5. Value of Gene Biomarkers in PD

Furthermore, the diagnostic efficacy of the genes was verified using the ROC curve in the validation dataset. Figure 7 indicates the specific AUC and 95% CI of the characteristic diagnostic genes: AGTR1 (AUC = 0.933), GBE1 (AUC = 0.967), TPBG (AUC = 0.767), and HSPA6 (AUC = 0.633). The results of such genes in the validation dataset were relatively satisfactory, which demonstrates powerful predictive capabilities.

### 3.6. Analysis of Immune Cell Infiltration

Immune infiltration of PD was computed using the CIBERSORT algorithm. The immune cell components of the disease and control groups can be seen in the bar chart (Figure 8A), and the differences between the same immune cells can also be seen in the violin graph (Figure 8B). Compared with the control group, the expression of B cell memory and dendritic cells (DCs) activation in the PD group was lower (*p* = 0.035 and *p* = 0.037, respectively) and that of M2 macrophages was higher (*p* = 0.024). Figure 8C indicates the interactions between the immune cells. Our results showed that B cell memory correlated positively with DCs activation (r = 0.42) and negatively with B cell initiation (r = −0.61). M2 macrophages correlated negatively with macrophages M0 (r = −0.58) but positively with monocytes (r = 0.33). The activated DCs had a significant negative correlation with B cells naive (r = −0.26).

### 3.7. Correlation Analysis between the Identified Genes and Immune Cell Infiltration

The correlation analysis of 22 types of immune cells could indicate how these identified genes take part in the development of PD by regulating immune cell infiltration. The results indicated that AGTR1 (R = −0.53, *p* = 0.0017), GBE1 (R = −0.38, *p* = 0.029), and TPBG (R = −0.46, *p* = 0.007) linked negatively to monocytes. AGTR1 related negatively to M2 macrophages (R = −0.46, *p* = 0.0073), and GBE1 related negatively to T cells CD4 memory resting (R = −0.35, *p* = 0.046), while HSPA6 related negatively to plasma cells (R = −0.45, *p* = 0.0089) (Figure 9).

## 4. Discussion

PD is the second most common neurodegenerative disease followed only by Alzheimer’s, the clinical assessment of which is usually tricky. It is often misdiagnosed because of the overlap of symptoms with other conditions [39]. The recent fast growth of bioinformatics has offered an effective solution for discovering and screening possible diagnostic genes. In the present study, 27 DEGs were screened from the PD expression profile extracted from GEO databases by differential analysis. Then, four PD-related hub genes, namely AGTR1, GBE1, TPBG, and HSPA6, were finally identified by LASSO and SVM-RFE algorithms. Later, these four hub genes were validated in an external dataset GSE20164. In addition, the CIBERSORT immune infiltration analysis revealed that these four hub genes were associated with increased infiltration of M2 macrophages, decreased infiltration of B cell memory, and activated DCs during the progression of PD.

Among the gene enrichment analysis of DEGs, the GO/KEGG enrichment analyses showed presynapse, synaptic vesicle, exocytic vesicle, neuron projection terminus, transport vesicle catecholamine binding, serotonergic synapse, and tyrosine metabolism and other immune-related signaling pathways. Tansey et al. demonstrated the causal role of inflammation and immune pathways in PD pathogenesis [40]. The DO enrichment analysis showed more clearly the association of PD with the occurrence of tumor inflammatory diseases, such as autonomic nervous system neoplasm, neuroblastoma, peripheral nervous system neoplasm, and synucleinopathy. This further implied the close relationship of PD to the immune response mechanism. Therefore, the data analyzed herein are of potential significance.

LASSO refers to a ML approach based on regression that can actively select from various potential multicollinear variables. We classified genes and variables by looking up the lambda parameter to find the smallest error [41]. Generally speaking, SVM is often viewed as one of the most salient and mature binary classification algorithms in microarray computing, especially useful for gene expression analysis [42]. We investigated AGTR1, GBE1, TPBG, and HSPA6, four specific genes, using the LASSO and SVM-RFE models and verified them using an external dataset.

The AGTR1 gene mediated by the renin–angiotensin system is crucial to the pathophysiology of cardiovascular diseases [43]. AGTR1 stimulation leads to physiological and pathological reactions, such as vasoconstriction, inflammation, and proliferation [44]. Moreover, studies in Japan have shown that AGTR1 gene variation is related to sporadic PD [45]. Notably, it is worth remarking that renin–angiotensin system inhibitors are a good solution for improving PD patients’ motor functions and reducing L-DOPA-related dyskinesia [46]. Moreover, the activation of AGTR1 in a PD mouse model was found to cause oxidative stress, leading to the loss of midbrain dopaminergic neurons, and its inhibition will prevent this [47]. The latest research shows that the single subtype characterized by the expression of the gene AGTR1 is limited to the ventral layer of the SN in terms of space, and it is highly sensitive to loss in PD, indicating a molecular process associated with degeneration [48]. In summary, such results indicate that AGTR1 may affect the selective susceptibility of dopaminergic neurons and that inhibitors of this pathway may affect neuroprotection.

The GBE1 gene belongs to the glycosyl hydrolase 13 family, whose mutations can cause adult polyglucosan body disease (APBD), which is a fatal adult-onset neurodegenerative disease featuring progressive sensory deficits and upper and lower motor neuron dysfunction [49]. In APBD, GBE is reduced and the glucan chains are too long, wind around each other, and roll up polyglucosan bodies (PBs), provoking neuroinflammation and neurodegeneration [50]. PBs formed in glia and neurons seem to clog, which may explain the neurological presentation of such a disease [51]. The low expression of GBE1 in patients with PD may also be related to neuroinflammation or neuronal tangle in the SN caused by BPs, which needs further study.

TPBG, an alias of Wnt-activated inhibitory factor 1 (WAIF1), refers to a single-pass transmembrane protein [52], which is usually highly expressed in trophoblast cells and tumors. Studies have found that it is also highly expressed in normal adult tissues, such as the brain, and TPBG is considered a PD-related gene [53]. The study has shown that gene ablation of TPBG causes slight degeneration of dopaminergic neurons in the midbrain of older mice. Furthermore, transcriptome analysis of the SN in older TPBG knockout mice confirmed TPBG as a potential candidate gene related to PD [54].

The HSPA6 gene encodes Hsp70B’, one of the stress-induced HSP70 proteins. There are no studies on the HSPA6 gene in animal models of neurodegenerative disorders in view of its sole existence in the human genome [55]. Previous studies found that the transcription activity of the stress-related HSPA6 gene was increased in PD patients in peripheral blood mononuclear cells [56]. Through bioinformatics research, we discovered that the HSPA6 gene is highly expressed in the SN of PD patients, with potential diagnostic value in PD recognition.

However, numerous studies are required to further prove the reliability of the diagnostic value of these genes. Recently, many studies have indicated the accelerating effect of immune cell infiltration into brain tissue on the disease process [57]. Therefore, the calculation of immune infiltration using the CIBERSORT algorithm is of great importance for discovering several immune subtypes closely related to the biological process of PD.

The increased infiltration of M2 macrophages and the decreased infiltration of B cells memory and activated dendritic cells (DCs) may be associated with the development of PD by way of neural injury and inflammation. Macrophages are the most significant innate immune cells in the brain as well as the most important regulators of neurodegenerative disorders. Some studies on multiple sclerosis have shown that activation of M2 macrophages/microglia can promote healing and repair [58]. Unfortunately, information on M2 macrophages/microglia markers is relatively lacking in PD or chronic animal models of PD based on α-synuclein. Our research seems to be in contrast to previous studies, but this also shows that the effect of immune response regarding M2 macrophages in PD is complex and needs to be further researched [59]. Researchers have proved that the number of B cells in PD patients may be reduced, including B cell memory [60]. Moreover, DCs turn out to be antigen-presenting cells in the pathogenesis of neuroinflammation. Studies have shown that immature DCs can activate endothelial cells more than activated DCs [61]. Our study implied the number of activated DCs in PD patients decreased, consistent with previous findings.

As for the correlation analysis, the gene biomarkers AGTR1, GBE1, and TPBG were all significantly related to infiltrating immune cells and monocytes. AGTR1 decreases the expression of M2 macrophages, GBE1 decreases the expression of T cells CD4 memory resting, and HSPA6 decreases the number of plasma cells. It can be noted that the pathophysiological mechanisms of PD include a large number of inflammatory cell changes and immune diseases. In the future, AGTR1, GBE1, TPBG, and HSPA6 could take part in the PD pathophysiological process through the role of such inflammatory and immune cells. In addition, the current studies on some PD immune targets add confidence to our further research, such as Wnt-related signaling [62] and G protein-coupled receptor-GPR109A [63], which may more precisely ameliorate neuroinflammation in diseases such as PD by controlling tissue or organ inflammation, thereby treating or delaying disease progression. The study also avoids the toxic side effects associated with the long-term use of drugs, such as levodopa. Our study also provides new ideas for PD immune-related treatments.

However, our research also has some limitations. Firstly, due to the fact of our limited sample size, a larger sample is needed for validation; secondly, our samples are from public databases and published literature. Further patient data from our research center can be collected for external validation, and multiple regression models can be used to validate the sensitivity and specificity of the selected biomarkers. Next, we will focus on the function of the screened potential genes and design complete in vivo and in vitro experiments for validation in the future.

## 5. Conclusions

In this work, we ultimately identified four hub genes, namely AGTR1, GBE1, TPBG, and HSPA6, by LASSO and support vector machine algorithms. In addition, we found that increased infiltration of M2 macrophages and decreased infiltration of B cell memory and activated DC may be involved in the progression of PD by way of neurological damage and inflammation, and AGTR1, GBE1 TPBG, and HSPA6 were significantly associated with the degree of immune cell infiltration. Our study may provide some insights into searching for potential immunotherapy targets for delaying or halting the progression of PD. However, more research is warranted to verify the role of these genes involved in the neuroimmune and neuroinflammation-related pathogenesis of PD in the future.

## Figures and Tables

**Figure 1 brainsci-13-00175-f001:**
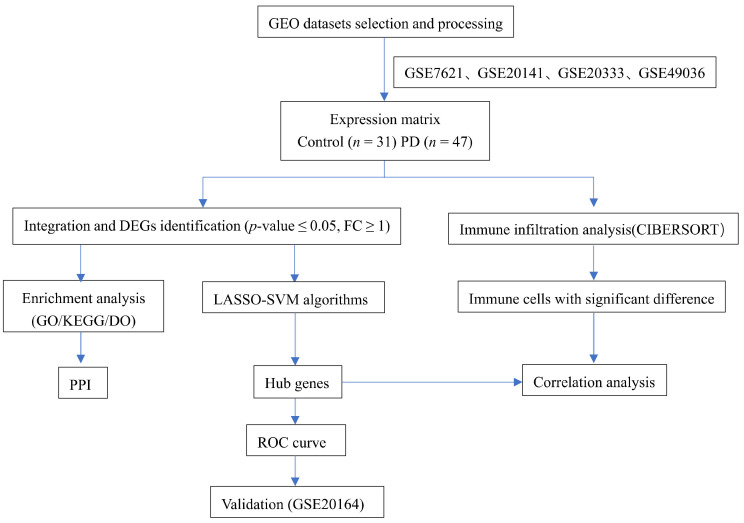
Flowchart of this study.

**Figure 2 brainsci-13-00175-f002:**
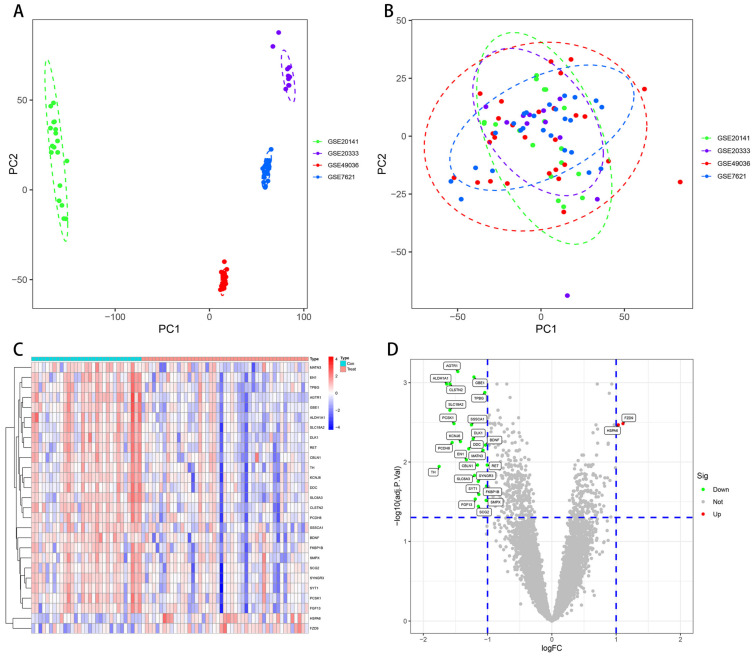
PCA and DEGs in substantia nigra between PD and normal controls. (**A**) pre-correction raw PCA; (**B**) post-correction combat PCA; and (**C**) heatmap indicating a significant DEGs. These two colors denote distinct trends; darker color for a more pronounced trend; (**D**) volcano map exhibiting DEGs. Red and green denote upregulated and downregulated genes, while grey denotes no significant difference. PCA: principal component analysis; DEGs: differentially expressed genes.

**Figure 3 brainsci-13-00175-f003:**
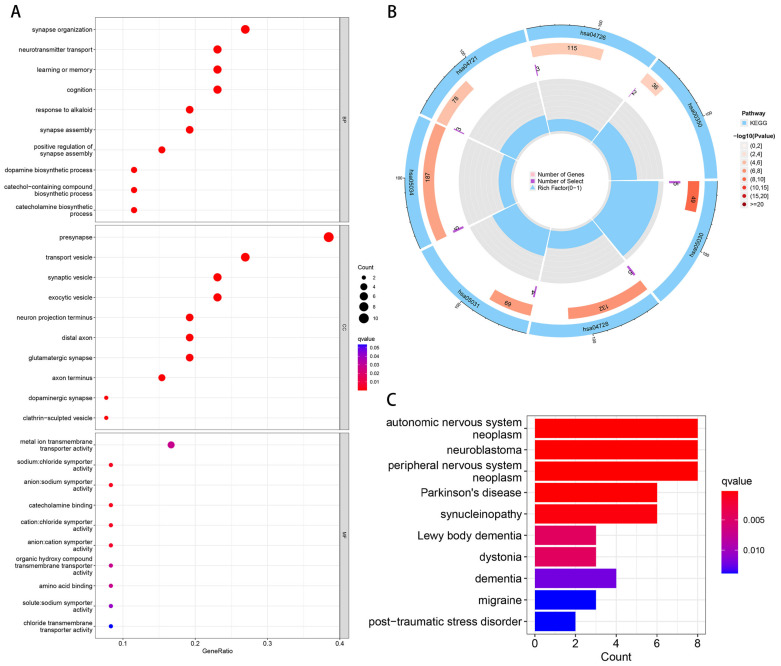
The results of the enrichment analysis of differential expression genes (DEGs). (**A**) Gene Ontology (GO) enrichment analysis, where the *x*-axis refers to the generation, and the *y*-axis refers to the significantly enriched GO analysis of the modules. (**B**) Kyoto Encyclopedia of Genes and Genomes (KEGG) enrichment using Circos plots. Each column in the outermost circle corresponds to a KEGG pathway. The second circle represents the number of genes contained in each pathway. The redder the color, the more significant the enrichment of DEGs. The third circle represents the number of DEGs enriched. The innermost circle represents the proportion of DEGs in the enriched genes of the pathway. (**C**) Disease Ontology (DO) enrichment analysis, where the *x*-axis refers to the gene count, and the *y*-axis refers to the enriched diseases.

**Figure 4 brainsci-13-00175-f004:**
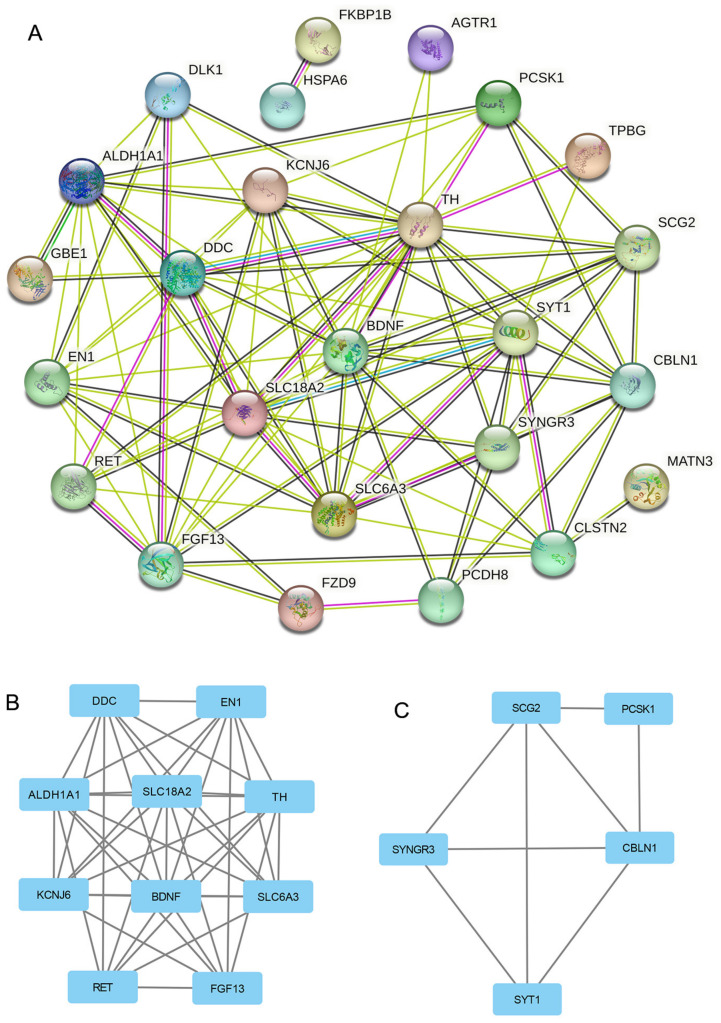
PPI network construction and 2 subcluster modules extracted by MCODE. (**A**) The interaction network among the proteins was coded by DEGs (25 nodes and 96 edges). The node refers to a protein, while the edges refer to protein–protein correlation between two nodes. (**B**) Subcluster module 1 was extracted by MCODE and consisted of 10 nodes and 41 edges; MCODE score = 9.111. (**C**) Subcluster module 2 consisted of 5 nodes and 8 edges; MCODE score = 4.

**Figure 5 brainsci-13-00175-f005:**
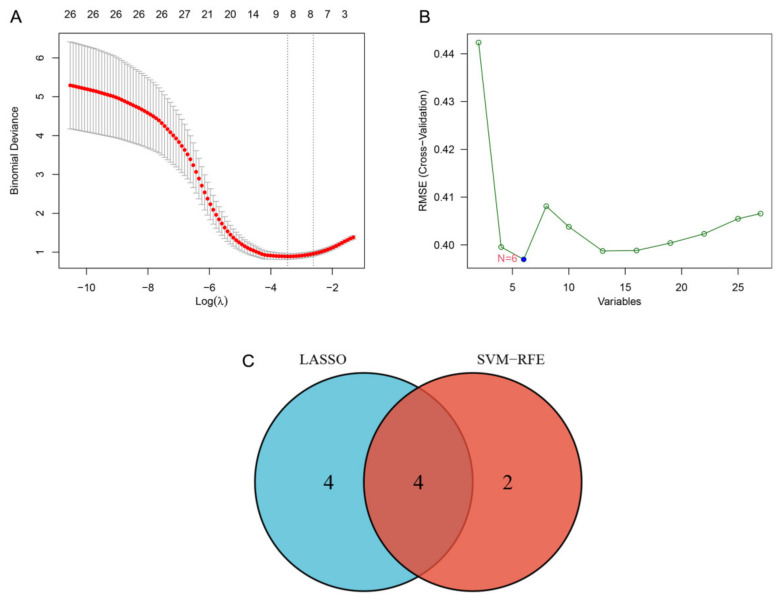
LASSO and SVM-RFE jointly screened and verified the special gene biomarkers. (**A**) eight genes were extracted for PD gene biomarkers with the LASSO algorithm; (**B**) six genes were extracted for PD gene biomarkers with the SVM-RFE algorithm; (**C**) Venn diagram indicating the four crossover genes between LASSO and SVM-RFE. LASSO: least absolute shrinkage and selection operator; SVM-RFE: support vector machine recursive feature elimination.

**Figure 6 brainsci-13-00175-f006:**
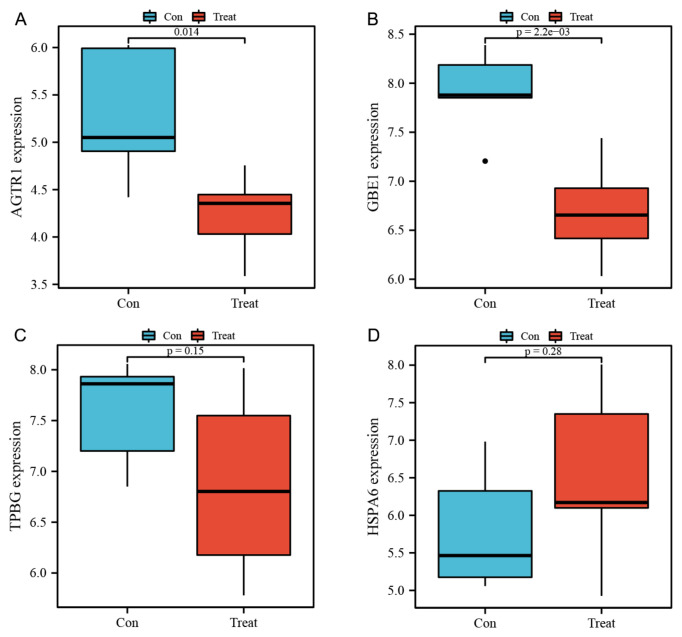
Expression levels of the four genes in the verification dataset GSE20164 for the substantia nigra samples of the control and PD groups. (**A**) AGTR: *p* = 0.014; (**B**) GBE1: *p* = 0.002; (**C**) TPBG: *p* = 0.15; (**D**) HSPA6: *p* = 0.28. *p* < 0.05 denotes statistical significance. AGTR1: angiotensin II type 1 receptor; GBE1: glycogen branching enzyme; TPBG: trophoblast glycoprotein; and HSPA6: heat shock 70-kDa protein 6.

**Figure 7 brainsci-13-00175-f007:**
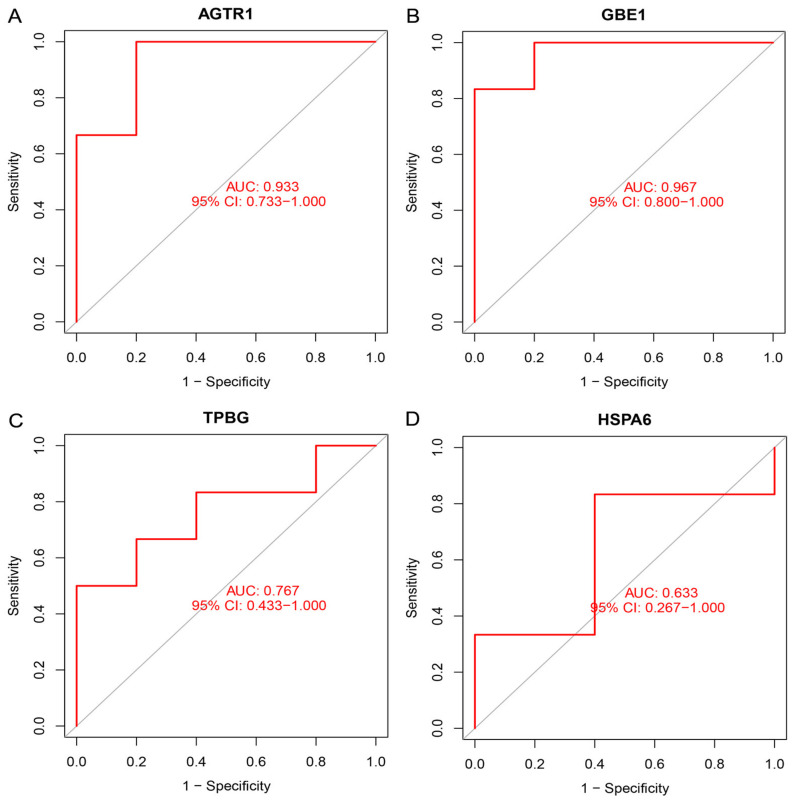
The ROC curves of four genes in the validation dataset. (**A**) AGTR: AUC = 0.933; (**B**) GBE1: AUC = 0.967; (**C**) TPBG: AUC = 0.767; and (**D**) HSPA6: AUC = 0.633. AGTR1: angiotensin II type 1 receptor; GBE1: glycogen branching enzyme; TPBG: trophoblast glycoprotein; and HSPA6: heat shock 70-kDa protein 6.

**Figure 8 brainsci-13-00175-f008:**
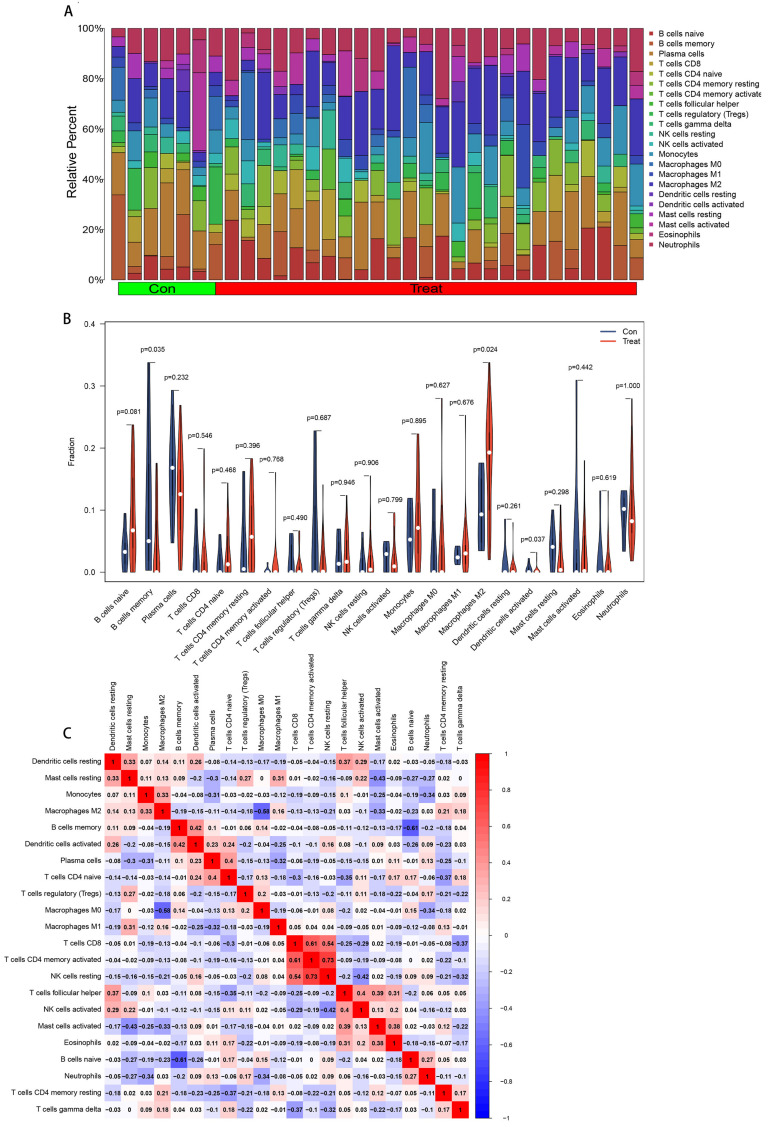
Analysis of the infiltrating immune cells. (**A**) The contrast of 22 types of immune cells’ proportion between the control group and treatment group. The *x*-axis refers to immune cells, and the *y*-axis refers to the relative percentage. (**B**) Discrepancy in the immune cell infiltration. Blue and red legends represent the control group vs. the PD group. The *x*-axis represents the type of immune cells, and the *y*-axis represents the fraction. *p* < 0.05 denotes statistical significance (B cells memory, M2 macrophages, and activated dendritic cells have significant differential infiltration). (**C**) Correlation in the immune cell infiltration. The *x*/*y*-axes represent the immune cell types, the red color refers to a positive correlation, and the blue refers to a negative correlation. Darker color represents a stronger association.

**Figure 9 brainsci-13-00175-f009:**
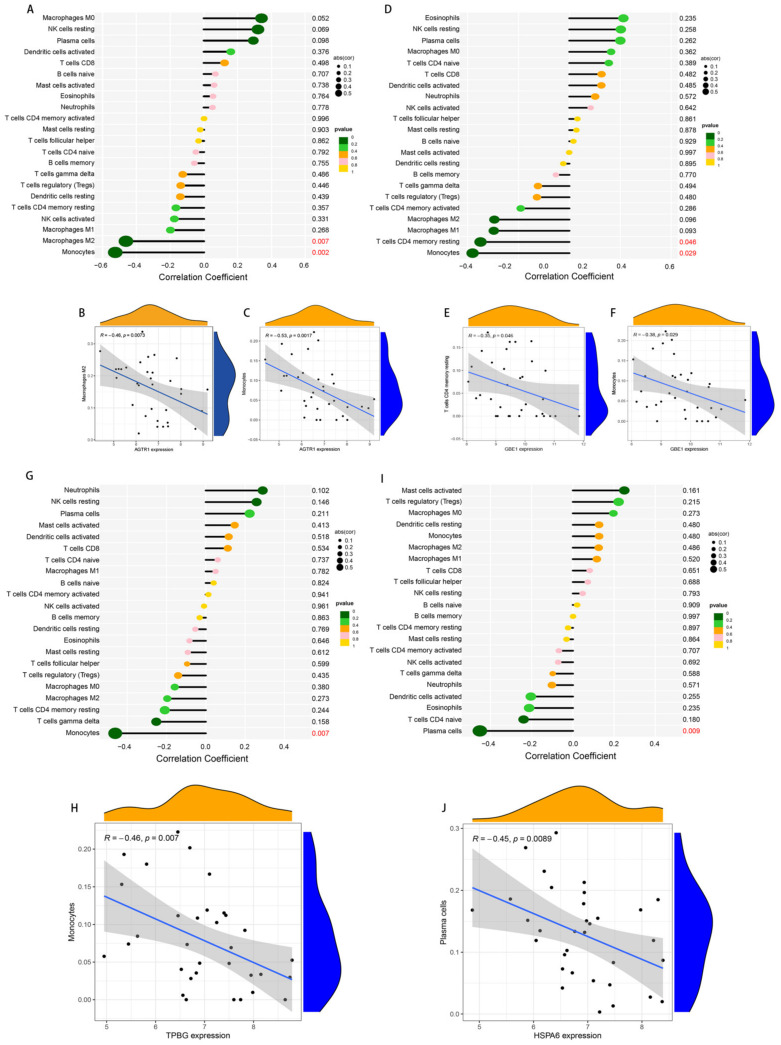
Immune cell infiltration correlations of the four selected genes. (**A**) lollipop plot of the correlation between AGTR and immune cells; (**B**,**C**) scatter plots of the significant correlation between AGTR and immune cells (M2 macrophages: R = −0.46, *p* = 0.0073; monocytes: R = −0.53, *p* = 0.0017); (**D**) lollipop plot of the correlation between GBE1 and immune cells; (**E**,**F**) scatter plots of the significant correlation between GBE1 and immune cells (T cells CD4 memory resting: R = −0.35, *p* = 0.046; monocytes: R = −0.38, *p* = 0.029); (**G**) lollipop plot of the correlation between TPBG and immune cells; (**H**) scatter plot of the significant correlation between TPBG and monocytes (R = −0.46, *p* = 0.007); (**I**) lollipop plot of the correlation between HSPA6 and immune cells; (**J**) scatter plot of the significant correlation between HSPA6 and plasma cells (R = −0.45, *p* = 0.0089). In the right column of lollipop plots, *p*-values < 0.05 with a red color indicate statistical significance. AGTR1: angiotensin II type 1 receptor; GBE1: glycogen branching enzyme; TPBG: trophoblast glycoprotein; and HSPA6: heat shock 70-kDa protein 6.

**Table 1 brainsci-13-00175-t001:** A summary of the PD datasets used in the analysis and independent validation.

Contributor	Accession	Platform	Samples (Normal/PD Sample)	Country	Last Update Date
Middleton FA	GSE20141	GPL570	8/10	USA	25 March 2019
Dijkstra AA	GSE49036	GPL570	8/15	The Netherlands	25 March 2019
Ffrench-Mullen JM	GSE7621	GPL570	9/16	USA	25 March 2019
Edna G	GSE20333	GPL201	6/6	Israel	25 October 2022
Hauser MA	GSE20164	GPL96	5/6	USA	10 August 2018

## Data Availability

All data related to the research are presented in the article.

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
