# Peer review of "Parkinson’s Disease Gene Biomarkers Screened by the LASSO and SVM Algorithms"

_brainsci, 2023, doi:10.3390/brainsci13020175_

Round 1

Reviewer 1 Report

Introduction

Broadly speaking, the introduction describes well and concisely what the focus of the paper is. In my opinion, more citations should be added that validate the positive effect of using immunosuppressants in PD. I would expand the state of the art on this very topic, which is certainly an important hook for the continuation of the discussion.

Add citation when you say “PD is usually diagnosed by physical examination and assessment of motor symptoms like resting tremor, muscle rigidity and bradykinesia (Line 31).

Materials and Methods

This section is very well structured and each part describes precisely the methods used. I would add some more information about LASSO (e.g., Scaling of the independent variables, grid/random search, tuning parameter λ about strength of penalty). I would extend the technical description about SVM-RFE for example by adding description of selected hyperparameter tuning (for further repeatability).

Results

This section is detailed, and the graphs accurately and consistently explain the results obtained. The plots are well done but those in Figure 8 a and b and 3 b are difficult to understand. I would keep the graph but expand the description in the caption.

Excellent Figures 5,6,7, which show in a clear and self-explanatory manner the result obtained as a result of gene selection.

Discussion and conclusion

The discussion about results is very broad and precise. The introductory cap (lines 208-211) is robust and well structured. The choice of discussing results by comparing with previous literature review is appreciable. By this method, the reader can directly make comparison.

I would extend conclusion section or integrate it inside discussion chapter. Moreover, when you say “Such gene biomarkers may be crucial in PD pathological mechanism or become therapeutic targets” (line 308), I would expand this concept with further arguments (e.g. why it can be crucial, what the alternatives to this study may be or which improvement might be done).

Author Response

We thank the reviewer for carefully reviewing our manuscript and providing important comments and suggestions. Overall, we agree with the comments/suggestions made by the reviewer regarding our manuscript, and these comments greatly helped us to improve the manuscript. And we have subjected our work to a professional language editing service as recommended to make our expressions more accurate. We hope that the reviewer will find our revised manuscript satisfactory. Please see the attachment.

Reviewer 2 Report

1.      Why screen potential Parkinson's disease genetic biomarkers? The problem behind the screening of Parkinson's disease genetic biomarkers with previous methods needs to be mentioned in brief.

2.      In the introduction section, the motivation, contribution, and organization of the study must be included for better readability. In addition to this, figure 1 is described the process for validating the selected potential genes, but under figure 1, the authors mentioned the caption as “flow chart of this study”.

3.      In section 2, the authors must improve the discussion in each sub-section or change the representation of the subheadings from numbering to bullet points as the content available in the sub-section is insufficient.

4.      The authors have submitted an incomplete article, where the conclusion is not written completely, and the references are not available in the article. Even the quality of the figures is very poor in the article.

Author Response

Thank you for reviewing our manuscript and providing valuable comments. The insightful comments/suggestions will greatly improve our study. It is a pity that did not described accurately or in detail enough in the previous version of the manuscript. In the revised version, we have addressed all the concerns raised by reviewers and we have subjected our work to a professional language editing service as recommended to make our expressions more accurate. We hope that the reviewer will find our point-by-point responses satisfactory. Please see the attachment.

Round 2

Reviewer 2 Report

The author has addressed the majority of the comments of the previous revision. However, there are a few changes that need to be incorporated for further improvement in the article.

Comments

1.     A comparative analysis needs to be included in the article, where the current study must compare with the existing studies for concluding the novelty and originality.

2.     In the introduction section, the motivation, contribution, and organization of the study are still missing in the article.

3.     The figures included in the article are very poor, their visibility is not clear, and difficult to understand what exactly the figure is conveying.

4.     In the article, a new section must be created to discuss the results of the study in brief before the conclusion section.

Author Response

We appreciate the reviewer for carefully reviewing our manuscript and providing constructive comments and suggestions again. We agree with the comments/suggestions made by the reviewer regarding our manuscript and revise the corresponding part according to the reviewer’s comments. We hope that the reviewer will find our revised manuscript satisfactory.Please see the attachment.
